# Solar Energy-Based Future Perspective for Organic Rankine Cycle Applications

**DOI:** 10.3390/mi13060944

**Published:** 2022-06-14

**Authors:** Raúl Alejandro Martínez-Sánchez, Juvenal Rodriguez-Resendiz, José Manuel Álvarez-Alvarado, Idalberto Macías-Socarrás

**Affiliations:** 1Facultad de Ingeniería, Universidad Autónoma de Querétaro, Querétaro 76010, Mexico; rmartinez109@alumnos.uaq.mx (R.A.M.-S.); jmalvarez@uaq.mx (J.M.Á.-A.); 2Facultad de Ciencias Agrarias, Departamento de Ciencias Agrarias, Universidad Estatal de la Península de Santa Elena, La Libertad 240204, Santa Elena, Ecuador; imacias@upse.edu.ec

**Keywords:** organic Rankine cycle, solar energy, photovoltaic cell, solar tower, parabolic dish, parabolic trough, linear fresnel reflector

## Abstract

This article explores the patents of solar energy technologies in the organic Rankine cycle (ORC) applications. The conversion of low-quality thermal energy into electricity is one of the main characteristics of an ORC, making efficient and viable technologies available today. However, only a few and outdated articles that analyze patents that use solar energy technologies in ORC applications exist. This leads to a lack of updated information regarding the number of published patents, International Patent Classification (IPC) codes associated with them, technology life cycle status, and the most relevant patented developments. Thus, this article conducts a current investigation of patents published between January 2010 and May 2022 using the Preferred Reporting Items for Systematic Reviews and Meta-Analyses (PRISMA) methodology and keywords. One thousand two hundred ninety-nine patents were obtained as part of the study and classified in F and Y groups of the IPC. The time-lapse analyzed was between January 2010 and May 2022. In 2014 and 2015, a peak of published patents was observed. China (CN) was the country that published the most significant number of patents worldwide. However, the European Patent Office (EP), the World Intellectual Property Organization (WO), and the United States (US) publish the patents with the highest number of patent citations. Furthermore, the possible trend regarding the development of patents for each technology is presented. A high-performance theoretical ORC plant based on the patent information analyzed by this article is introduced. Finally, exploration of IPC revealed 17 codes related to solar energy technologies in ORC applications not indexed in the main search.

## 1. Introduction

For business policy making groups, legal and business departments, and industries, it is essential to have a database of articles that provide information about existing patents in their branches of operation [1]. Four fundamental variables are obtained from the analysis of patents [2]:Competitor analysis;Testing and monitoring technology;Mastery of crucial technology;Identification of trends and conditions for the development of patents in different markets.

The constant search for trends in technological developments enables managers and decision-makers to identify innovations in the different branches of interest. This is important for research and development (R&D) in the government and business sectors, since it establishes strategies for identifying and monitoring trends in developing new technologies from a technical perspective [3].

On a global level, companies are in the process of constant improvement to be sustainable in the long term. Business competition is fierce, and introducing innovations or promoting R&D activities can make a difference on the road to success. Patent development can be considered a rectangular stone to gain a competitive advantage [4].

The annual number of patent applications worldwide has been increasing. This has made the ability to process all applications a cumbersome process for the patent examiners [5]. The in-depth knowledge of the terminology and structure of IPC allows for searching and analyzing patents with accuracy since it covers the spectrum where a specific patent can be indexed [6].

A review of the status of published patents on using the solar resource for implementation in an ORC is vital for developing current and future research. Moreover, it helps to determine the trends of scientific research lines and the status of the different technologies worldwide. It will also enable researchers to understand if the product they are working on has been developed elsewhere [7].

The objective of this article is to explore the status of patents on the application of solar energy in an ORC. In the last decade, technological advances increased solar collection efficiency. Due to these factors, new techniques have been implemented to supply energy to an ORC [8]. Therefore, many patents and applications worldwide have been generated. Knowing if a product has been developed saves manufacturing time and financial resources and makes it easier to direct the research efforts in a different direction [9].

The article is structured as follows: Section 1, introduces the topic, its importance, and the contribution it intends to achieve. Section 2 presents the current context for the development of the article. Section 3 defines the keywords and the methodology to obtain the analyzed information. Section 4 discusses the results obtained by analyzing tables and figures. Finally, Section 5 presents the conclusions, bibliographical references, terminologies used and future perspectives are shown.

## 2. Background

An ORC is a thermodynamic system that allows to produce power or work and behaves in the same way as a conventional Rankine steam cycle. The main difference is in the working fluid and the operating temperature of the cycle [10].

ORC-based systems are a viable option for power generation as it is easier to obtain the necessary components for their operation on the market [11]. The ORC consists of four stages [12]:Isentropic compression in the pump;Isobaric evaporation in the boiler;Isentropic expansion in the turbine;Isobaric condensation in the condenser.

As for the working fluids, the ORC uses organic refrigerants with low condensation and boiling points. This withstands the use of low-grade heat, low temperature, and pressure [13].

An ORC plant can use heat sources of up to 400 °C and generate electrical power in a range of 1 kW and 10 MW [14]. The implementation of solar collection is a sustainable way of supplying energy carriers due to the compatibility with the operating temperature of the ORC [15].

Developing and implementing ORC-based and solar collection systems is a way to generate electricity in an environmentally friendly way. This strategy reduces greenhouse gases in the atmosphere, and sustainable technology to produce electricity is standardized [16].

The sun is the natural regulator of temperature in the terrestrial atmosphere. Daily, it radiates to our planet the energy consumed by society in one year. Electricity production from this type of renewable energy has increased due to its potential and the technological advances made in capturing solar radiation, as shown in Figure 1 [17].

Different technologies can convert solar energy into electricity or heat by using a solar tower, a photovoltaic cell, a parabolic dish, p and a linear Fresnel reflector [19]. Solar radiation is a free and inexhaustible energy. The technologies used for its operation have a durability of more than 15 years [20].

ORC plants are not designed to use solar energy. Due to the development of organic fluids and the increasing efficiency in solar collection, it has been possible to incorporate them into its operation [21]. Traditionally, ORC has been combined with other heat sources such as engine exhaust gases, boilers, etc. [22]. The emission reduction potential in the high proportion renewable energy system for ORC applications should be explored due to its value in the trading market [23].

Thus, Figure 2 presents five main types of technologies for power generation using solar energy technologies in ORC applications.

## 3. Theoretical Bases

Keywords are useful for indexing large volumes of information from the web. However, refining the search methodology is vital to obtaining accurate results [24]. Today there are more than 100 offices and 120 million patents worldwide. All this information is available on the internet. Therefore, performing an efficient search could reduce R&D times [25].

Table 1 depicts the search strategy used to obtain the list of patents related to implementing solar collection technologies in an ORC.

Technology research and development strategies can be established by consulting the existing patents for technology such as educational institutions, legal entities, and industries. Additionally, the respective economic incentive that entails the commercialization of a patented product or technology [26].

The patent status, date of publication, country, and name of the inventor or assignee is found on the internet. However, to obtain accurate information, search criteria and reliable query should be established [27].

The information is disaggregated in the search for published patents referring to the use of solar energy in an ORC. Likewise, obtaining the IPC code where a patent is classified is complex. In some cases, patents are shown that are not related to the search that has been defined. A study that organizes the information of the number of patents by technology, year of publication, relevant developments, IPC codes where they are indexed, and countries or assignees that are at the forefront in the publication of patents, is necessary to establish the degree of penetration of power generation systems that use an ORC in different countries. Establishing cooperative alliances to promote its use where conditions encourage the development of sustainable technologies [28,29].

The indexed results are compared using the exact keywords in Google Patents and Scopus Patent. However, when analyzing the data obtained using Excel tools based on the names of the patents, IPC code, country/assignee, patent date publication, and inventor name, it is noted that the indexing carried out by Google Patents contains all results returned by Scopus Patent. Therefore, Google Patent was chosen as the default search engine.

Figure 3 displays the patent selection process for this article using the PRISMA methodology [30]. The time-lapsed analyzed is between January 2010 and May 2022.

Patents to encounter the eligibility criteria meet specific requirements:The latest version of the patent must be published no earlier than 1 January 2010, and no later than 1 May 2022;The name of the patent cannot be registered in more than one country or assignee the name of a patent;Patents should be related to parabolic dish, parabolic through, photovoltaic cell, solar tower, and linear Fresnel reflector technologies in an ORC.

As part of the eligibility process established in the Prisma methodology, there is no duplication in the name of the patents so that the total number of patents to be shown always refers to innovations that are different from each other.

Search settings include indexing in all languages, although the results are displayed in English.

Of the initial sample of 3510 patents, 743 were removed due to duplicate names. One thousand two hundred sixty-eight patents were eliminated because they were not directly related to an ORC. Additionally, 200 were terminated for filing active claims. Thus, 1299 patents met the eligibility criteria and are part of the article.

## 4. Results and Discussion

In this section, the analysis results are discussed.

Figure 4 schematizes the distribution of patents according to each technology. For example, PVC and STP are solar collector systems with the highest number of published patents with 35.7% and 25.6%, respectively. Then, there is PTC with 15.2%, LFR with 12.1%, and PDC with 11.5%. The last three technologies present a similar number of developments published in the analyzed period.

The number of patents published on PVC and STP could be given by the cost of the equipment used to put this type of technology into operation, energy efficiency, and the durability of the devices [31].

Figure 5 shows the evolution of the number of patents published between 2010 and 2022. There is an average of similar publications from the beginning of the period until 2012.

A considerable increase in the number of patents occurred during 2013. The maximum peak was reached in 2014 with 219 published patents. The growing disclosure of the benefits and facilities offered by an ORC allows for an increase in the number of published patents [27].

The number of publications decreased moderately in 2015. However, in 2016 there was a considerable rise in the number of published patents, and between 2017 and 2019, there was a similar average in patented developments. In 2020 and 2021, there was a downward trend, with only 81 and 44 patents being published yearly, respectively. Until 1 May 2022, six patents had been published.

These figures are alarming for developing plants with ORC and incorporating PVC, STP, LFR, PDC, and PTC systems into their operation. The COVID-19 pandemic, which has paralyzed R&D activities, and the loss of interest in generating energy with solar technology in ORC applications, could be fundamental factors for the decrease of published patents [32].

Annual average of patents published from 2010 to 2022 was 108. Therefore, monitoring number of patents issued in the period 2022–2030 will give insight into the technological progress in an ORC through solar collection systems.

Figure 6 shows countries and organizations with the most significant impact on patent publications worldwide. China (CN) is the largest generator of publications, with a third of the generated patents. However, the United States (US), the World Intellectual Property Organization (WO), and the European Patent Office (EP) also have a significant number of patent publications. Developed countries or their assignees have more than 92% of published patents.

Of the countries in the Americas, excluding the English-speaking countries, only Brazil and Mexico have patent publications; 31 and 6, respectively. Central America and part of South America have good solar radiation values to implement ORC plants that incorporate solar collection systems [33].

Figure 7 sorts worldwide published patents for solar energy technologies in ORC application by country. CN has the most significant number of publications, with 468. 39.7% belongs to PVC, 32.1% to STP, 13.5% to PDC, 10.3% to PTC, and 4.4% to LFR.

CN has promoted international scientific collaboration, which has favored the development of its researchers and science and technology. Implementing government programs to develop technology and construction of large factories for firms from other countries have provided the opportunity to perform the latest scientific advances first-hand [34].

US, WO, and EP have similar numbers of published patents for solar energy technologies in ORC applications. Although CN has most significant number of publications, the US has the largest number of citations per capita, setting the trend in development lines [35].

Data analysis obtained in the tables demonstrates how CN dramatically impacts the number of patent publications. As long as solar energy technologies in ORC applications are considered necessary, the percentage of publications worldwide will remain stable. However, the US has the most relevant patents published between 2010 and 2022, which is synonymous with the quality of the research they generate. Other offices, such as WO and EP, also have patents of interest that seem helpful in improving the performance of ORC applications. The almost non-existent number of patents developed in Central and South America depicts how far the region is from being part of the select group of inventors and developers of solar energy technologies. The short distance from a a world power, such as the US, should enable the search for cooperation mechanisms, in order to promote and massify the use of the ORC application for power generation.

### 4.1. Future Perspective for Solar Energy Technologies in Orc Applications

Figure 8 exhibits the trend of patents for each technology described by this study. The number of publications grew steadily from 2010 to 2014. From 2015 on, the number of publications starts decreasing; although it was not until 2019 that the number of published patents fell abruptly to levels below those averaged up to that date. This indicates that the developments of ORC applications worldwide have suffered a recession. However, when analyzing the publications, it can be seen how PVC technology is gaining ground as the system with the most patents per capita yearly.

The information in Figure 8 raises an unknown about the future of solar energy technologies in ORC applications. The results indicate that PVC systems have the most significant potential for future updates and developments. However, attention should be paid to all the technologies involved in collecting solar radiation because a novel technical or material may be developed shortly and thus reverse the current trend.

### 4.2. Patent Family

Based on the strategy defined by the assignee, a patent can be filed in one or several countries. The first patent application is considered the main application. A patent family is generally defined as a group of patents usually filed in several countries/offices to protect the innovation from possible plagiarism. However, there are some issues regarding the meaning of patent family.

Data on patent families are most useful for the topics described below [36]:To prevent double counting of one invention;To suffer from a home bias and overestimate the patent propensity of residents because applicants are more likely to apply in their home country first;To forecast the number of patent applications to plan future resource requirements at patent offices;To analyze the internationalization of technology markets;To study the economic value of patents, as well as the strategies employed by applicants.

Two of the most widely used definitions of patent family are detailed below. First, there are the equivalent patents, which identify patents that protect the same invention. According to the EP statutes [37], they are defined as: “All documents with the same priority or combination of priorities belong to one patent family”.

The second definition refers to the extended patent families, and according to The International Patent Documentation Center (INPADOC): “All the documents directly or indirectly linked via a priority document belonging to one patent family” [38]. This article applied equivalent patent definition to analyze the patent data. refers exclusively to the main application and priority office.

### 4.3. International Patent Classification

On 28 September 1979, the IPC was established by signing the Strasbourg Agreement. According to the different areas of existing technology, there is a hierarchy system through symbols without the use of language [39].

The IPC classifies technologies into eight groups and divides them into more than 70,000 subgroups. Each subgroup consists of a symbol that consists of Arabic numerals and letters of the Latin alphabet. This system is essential for the organization and searches for patents, as well as for knowing the state of the art of these possible similar developments, which is useful for people or companies interested in developing a specific technology. A new version of the IPC comes into force every 1 January [40].

Figure 9 displays the IPC codes present in patents selected for this review. As can be seen, 77% of the patents are in group F. According to the EP, this group is called mechanical engineering, lighting, heating, weapons as well as blasting [41].

The remaining 23% is concentrated in group Y. As mentioned by EP [41], group Y is called general tagging of new technological developments; general tagging of cross-sectional technologies spanning over several sections of the IPC; technical subjects covered by former USPC cross-reference art collections [XRACs] and digests. Knowing the categories within the IPC makes it possible to quickly find patents related to specific technologies.

Table 2 presents a detailed description of the IPC codes that are part of the article. With the analysis of Table 2, it is easier to understand where the patents are indexed in the IPC.

Having the IPC code at hand may not be feasible, and it is well-known that it is difficult to memorize the codes. In fact, the discussion derived from Table 2, determines that there are patent codes on solar energy technologies in ORC applications that have not been described so far.

Table 3 presents the IPC codes linked to solar energy technologies in ORC applications that were not included in the initial search. The lack of precision in the indexed data happens when only keywords are used when performing the search. As a rule, concepts associated with and specific to a patent are not used when searching for information. Likewise, IPC codes are not tracked in the search for patent information. The consequence is the appearance of patents that are not within the defined search range [44].

The most important result of the article is the non-appearance of the IPC codes related to solar energies technologies in ORC applications in the keyword search is shown in Table 3. The analysis of studies such as [44] makes it possible to establish a procedure for correctly indexing of published patents regarding the subject of study. The conclusion is that with the sole use of keywords, results obtained in the search only show 81% of the total number of patents. Therefore, it is necessary to modify the methodology. First, the search is carried out by keywords. Then, the search for concepts specific to the group of patents is performed. Finally, the IPC codes where the patents in question can be indexed is establsihed. In the case of the search for solar energy technologies in ORC applications, 17 IPC codes were not initially in the group of patents selected for the study. Knowing the scope of a group and its sections is crucial to classifying a patent correctly.

Advanced knowledge of the groups and subsections where patents can be found in a particular technology permits the inventor/assignee to know if his development can be patented. Complementing the patent search through keywords with the help of IPC codes becomes a valuable tool to obtain accurate results immediately [48].

### 4.4. Technology Updates

Some current relevant patents on solar energy technologies in ORC applications are presented in this section. Technology upgrades in each category are clarified below.

#### 4.4.1. Parabolic Dish

The innovation in [49] presents an engine motor plant that uses renewable energy. The feeding system consists of a parabolic dish system for solar radiant heat collecting, a device to guide solar energy to a heating chamber, and a biomass processing system. Through the processing of biomass, thermal energy is generated. A fuel fluid processing system produces thermal energy through the combustion of the fuel fluid inside the combustion chamber. On the other hand, a closed-cycle thermodynamic based engine is powered by the solar energy captured by the parabolic dish system and the thermal energy obtained from the combustion chamber. This leads to the operation of the thermodynamic engine producing electrical energy.

#### 4.4.2. Parabolic Through

The patent [50] presents a system that enables rapid reheating of a thermal fluid through a reflective compensated parabolic trough and solar concentrator in a longitudinal greenhouse box. In this way, hot water and medium temperature steam are obtained. A greenhouse box is one of the preferred devices to generate steam and hot water in large volumes since it has better solar thermal captures. The automation system necessary to control the flows employs solenoid valves, pressures, and temperatures using pressure switches and thermostats. In this way, the latest generation heat pumps allow the transfer of fluids in a constant and fast way, regardless of the time of day or if the time in terms of the abundance of solar radiation is not optimal.

The thermetic longitudinal greenhouse box is installed at a designated reflecting point and from where both direct and upper radiation is captured. The already high temperature provided by the lower reflection is concentrated by the compensated paraboloid gutter (temperature up to 70 times more than standard surfaces). The design facilitates rapid reheating of large volumes of thermal fluid.

#### 4.4.3. Solar Tower

The innovation in [51] features a biomass energy-tower type solar energy coupling power generation system based on an ORC. The system comprises an ORC for power generation, a solar collection system, and an energy system based on the use of biomass. The solar collection system has a group of heliostats that reflect solar radiation on the solar tower, and through this process, thermal energy is generated. The energy generation system is made up of a boiler from which the caloric power is generated to heat the biomass through biomass. A storage tank supplies the boiler with water. The ORC system has a heat exchanger, an electric generator coupled to the turbine, a condenser, and a pump. The described system combines energy from biomass with solar energy obtained through the solar tower system. In this way, the utilization index of the solar resource and the thermal efficiency in the circulation of the fluid within the cycle is improved. Innovation solves the problems related to the solar collection system since different energy generation techniques based on renewable sources are embedded, an off-grid operation is possible 24 h a day.

#### 4.4.4. Linear Fresnel Reflector

The patent [52] presents a design to generate energy and desalinate seawater; this is achieved by concentrating solar energy with other cogeneration techniques. A solar collection system based on an inclined Fresnel mirror is used from which solar radiation is directed towards a central receiving tube. In this way, the temperature of the working fluid that circulates inside the receiver tube is raised, thus producing saturated or wet-dry steam, which, as it passes through the rotary screw expander motor, causes the rotation of the shaft coupled to a generator. The residual heat of the working fluid is used in the seawater desalination process through a Multi-Effect-Distillation system. When solar energy is unavailable during production hours, a molten-salts Single-Temperature-Thermal-Energy storage system is used as the battery. The residual heat from the industrial process and the solar collection system generates electrical energy through an ORC by applying caloric power to a working fluid, which is used to desalinate seawater.

#### 4.4.5. Photovoltaic Cell

The technology [53] describes a method of cooling solar panels while recovering energy. The system comprises a photovoltaic panel designed to transform part of the solar radiation captured into electrical energy. Inside the system, a circuit is installed with a heat exchanger configured to circulate a working fluid that extracts part of the absorbed heat in its path near the solar panel. The innovation is designed in a cascade to transfer enthalpy from the first working fluid to a second working fluid. The thermal energy of the second fluid passes through a turbine coupled to an electric generator.

### 4.5. Most Frequently Cited Patents

The cited patents represent the most outstanding technological advances published between 2010 and 2022. During these years, solar energy has grown due to the increase in the efficiency of solar capture [54]. First, however, it is necessary to highlight some milestones in developing solar energy technology in ORC applications. Many citations presuppose that the theoretical value, practice, and importance are more remarkable than other patents [2]. Furthermore, it establishes that the possibility of obtaining profits and the economic value of the patent are superior [55,56]. The number patent citations is a source of information commonly used when consulting the literature on innovation. A patent is positively linked to the quality of the patented innovation [57].

Table 4 exhibits the number of times a specific patent was cited in another patent. Patent citation articles such as those in journals such as the Science Citation Index (SCI) are outside the scope of this study. However, it is surprising that only 1.5% of patents generated in the US are cited in SCI journals [58].

Aristodemou and Tietze [59] posit the existence of nine forward citation-based measures, which are the most representative indicators of technological impact. The parameters are divided into two groups. The first group are the relevant characteristics at the patent level (citation index, forward citation frequency, generality, influence). The second group has relevant characteristics regarding the patent portfolio level (current impact index, Herfindahl–Hirschman index, hindrance index, relative patent position, technology strength).

The authors [60], argue that the number of citations of innovation has been used for decades for empirical economic analysis. They dispute that in 1995 it was likely that a published patent would offer more information than one issued in 2015. They also suggest that changes in the patent generation processes and current search methodologies, in many cases, lead to results indexed as invalid or biased.

Table 4 summarizes the most relevant patents on solar energy technologies in ORC applications. This allows us to concisely analyze the cited patents and see which ones can be incorporated into an ongoing investigation; thus enabling the reduction of R&D times and its benefits. Implementation of the cited patents could improve the performance of an ORC plant, provide an effective solution for an ongoing project or lay the theoretical foundations for the development of future studies in the field. The analysis of the patents published establishes a pattern regarding the trends regarding in developing solar technologies for ORC applications. Completing registering a patent is a long process, with the elaboration of technical documentation and administrative mechanisms. However, to reach this step, the nature of the invention must have been analyzed and detailed beforehand. Patents are success stories that help technological development at a global level. Using them as development tools can allow to find new techniques.

**Table 4 micromachines-13-00944-t004:** Patent updates for solar energy technologies in ORC applications.

Patent Title/Date	Patent Number	Inventor/Assignee	IPC	Times Cited
Hybrid solar/non-solar energy generation system and method/2013	WO2013059112A1	Jonathan Falcey [61]/WIPO	F03G6/06	29
Heat pipe type solar energy ORC low-tempe rature thermal power generating system/2012	CN101761461B	Jie et al. [62]/CN	Y02E10/44	21
Organic rankine cycle for concentrated solar power system/2014	KR20140015422A	Kosamana et al. [63]/KR	Y02E10/46	21
Organic rankine cycle for concentrated solar power system with saturated liquid storage and method/2014	MX2013011348A	Kosamana et al. [64]/MX	F03G6/003	7
Solar photothermal combined power generation system/2014	CN106321382A	Yanping et al. [65]/CN	F03G6/061	12
Integrated cascading cycle solar thermal plants/2020	US10690121B2	Yogi et al. [66]/US	F03G6/065	19
A hybrid photovoltaic system and method thereof/2011	EP2398070A2	Chatterjee et al. [67]/EP	Y02E10/50	20
Hybrid thermal power and desalination apparatus and methods/2018	US9932970B1	Donald Jeter [68]/US	Y02E10/46	14
Supercritical carbon dioxide power cycle configuration for use in concentrating solar power systems/2016	US20120216536A1	Ma et al. [69]/US	F03G6/00	12
Solar power plant/2014	US8661778B2	Bronicki et al. [70]/US	F03G6/066	6
Renewable energy storage system/2015	EP2836769A2	Dearman et al. [71]/EP	F03G6/003	23
Photovoltaic-thermal solar energy collection system with energy storage/2016	US20160156309A1	Almogy et al. [72]/US	F03G6/001	8
Photovoltaic-thermal solar energy collection system with energy storage/2016	US20160156309A1	Almogy et al. [72]/US	F03G6/001	8
Steam turbine plant/2018	EP2846008B1	Goto et al. [73]/EP	Y02E10/46	32
Organic rankine cycle decompression heat engine/2019	US10400635B2	Johnson et al. [74]/US	Y02E10/50	20
Hybrid solar concentration device/2012	EP2518781A2	Chatterjee et al./ EP	Y02E10/50	10
Improved brayton photothermal power generation method and system/2019	WO2019000941A1	Zhiyong et al. [75]/WIPO	F03G6/063	8
Hydroelectric solar tower with punctual concentration/2017	FR3025593A1	Kheir Mazri [76]/FR	F03G6/06	5
Solar thermal power generation facility/2021	US11060424B2	Umaya et al. [77]/US	F03G6/00	16
Thermal energy storage and retrieval systems/2017	US9845998B2	Sten Kreuger [78]/US	F03G6/003	26

### 4.6. High Performance Theoretical Orc Plant

This section proposes an ideal theoretical configuration from patents to achieve a high-performance ORC plant using solar energy technologies. In ORC applications, some factors can alter its performance; working organic fluid, solar collection system, and turbine. For each factor, a theoretical solution is proposed below.

The invention is related to a working fluid designed to obtain a greater cycle and general efficiency of the ORC system and reduce the impact of gases on climate change. The novel working fluid has a chemical composition of 4-hexafluoro-2-butene (HFO-1336mzz-E) or a mixture thereof. IPC codes for this patent are F01K25/08 [79]. The invention uses a collimated or otherwise concentrated beam of solar radiation to heat a ceramic device with a high absorption capacity. This device transmits the heat to a thermal storage unit by conduction. IPC codes for this patent are F03G6/065 [80]. The invention describes the calculation model for the difference in quantitative consumption for the turbine operation under variable pressure conditions. The main characteristic variables are the effective enthalpy drop of a high-pressure cylinder and the enthalpy rise of a water feed pump of 1 kg of unit steam and adopt an enthalpy drop correction coefficient. IPC codes for this patent are G06F30/20 [81].

## 5. Conclusions

This article has analyzed published patents referring to the use of solar energy technologies in ORC applications. The search engine used to obtain patent data was Google Patents. The analysis of the patents was obtained after applying the PRISMA methodology to obtain the quantity to be analyzed. The findings obtained from this study are listed below.

The number of published patents grew gradually from 2010 to 2014 when the maximum number of publications was reached. However, as of 2015, there is a slight downward trend. Starting in 2020, this phenomenon has been accentuated, and so far, in 2022, the number of published patents is lower than the previous annual averages;Two groups of the IPC contain patents related to solar energy technologies in ORC applications. Group F owns 77% and group Y 23% of the total number of patents, respectively;Although groups F and G contain the patents related to the topic of the article, when performing the search, only part of the IPC codes was indexed. Seventeen IPC codes that belong to patents related to solar energy technologies in ORC applications were collected;TheUS, WO, and EP were the countries/assignees with the most significant impact due to the number of citations of their patents. Although a predominant number of publications come from CN, this indicates that there is a prevailing number of studies and potential for future development;Patents related to Photovoltaic cell technology present the most stable number of patent publications per year as data;Patents that will possibly implement a high-performance theoretical ORC plant are filed.

Studies to be carried out in the future should focus on the evolution of PVC and STP technologies, expecting new developments to be improved. These would enable the identification of new horizons for the development of patents and also the evaluation of R&D trends to identify in which areas innovations are expected to develop.

## Figures and Tables

**Figure 1 micromachines-13-00944-f001:**
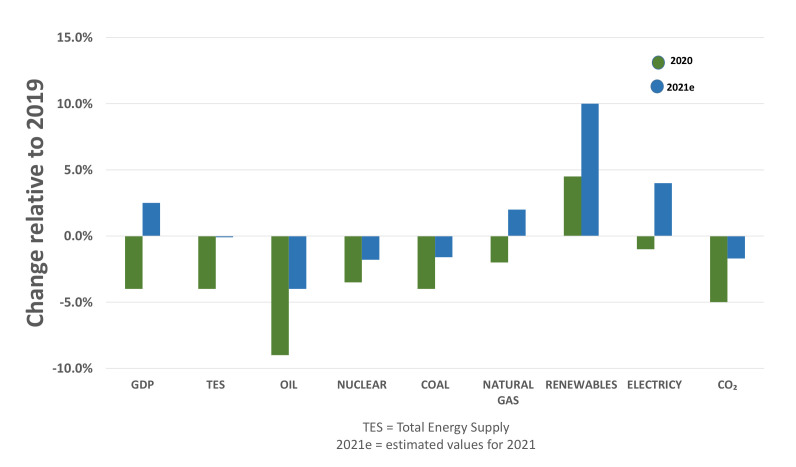
Change in key global indicators for energy demand and emissions, 2020 and 2021. Reprinted from Ref. [18].

**Figure 2 micromachines-13-00944-f002:**
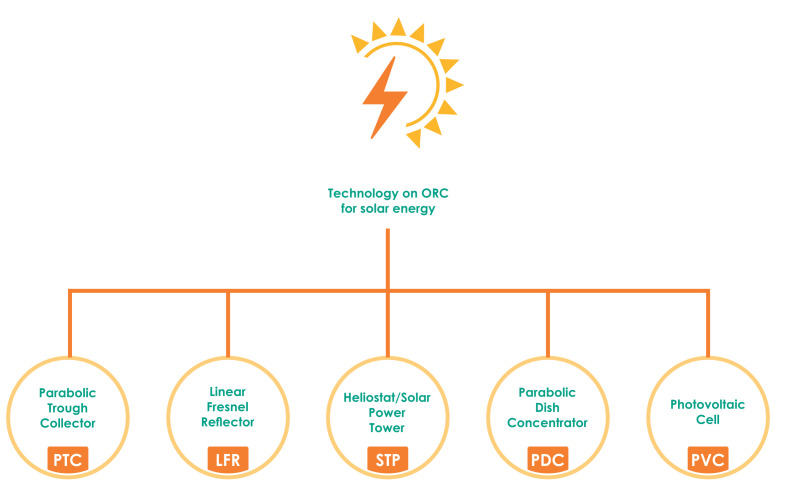
Technologies of ORC application for solar energy.

**Figure 3 micromachines-13-00944-f003:**
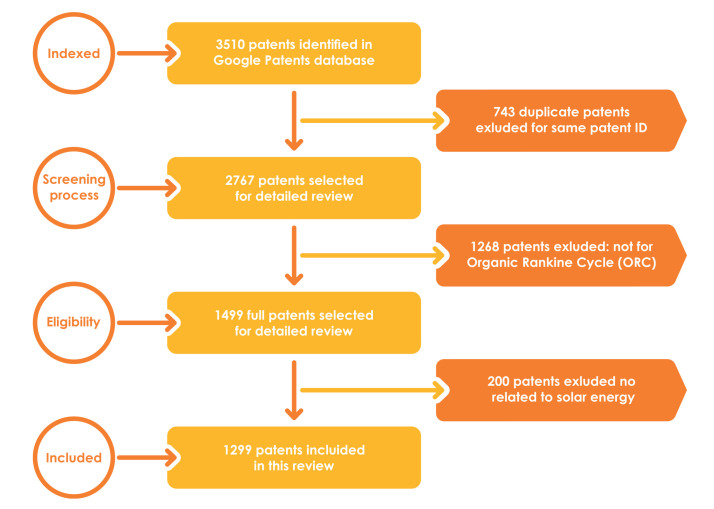
Process adopted to review patents results based on PRISMA.

**Figure 4 micromachines-13-00944-f004:**
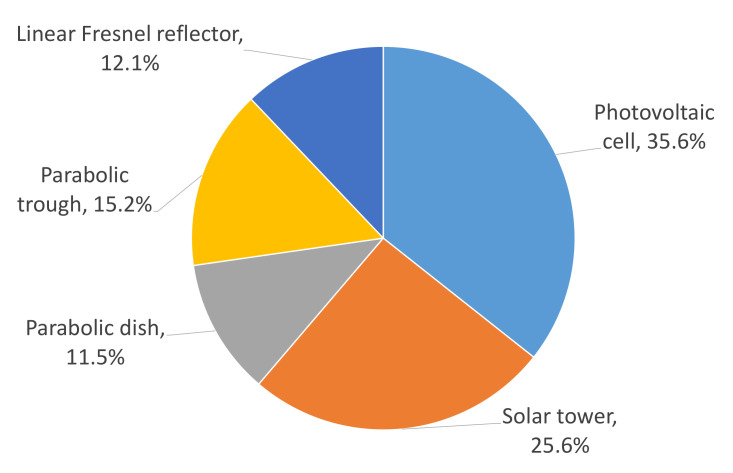
Distribution of different technologies in ORC applications for solar energy patents.

**Figure 5 micromachines-13-00944-f005:**
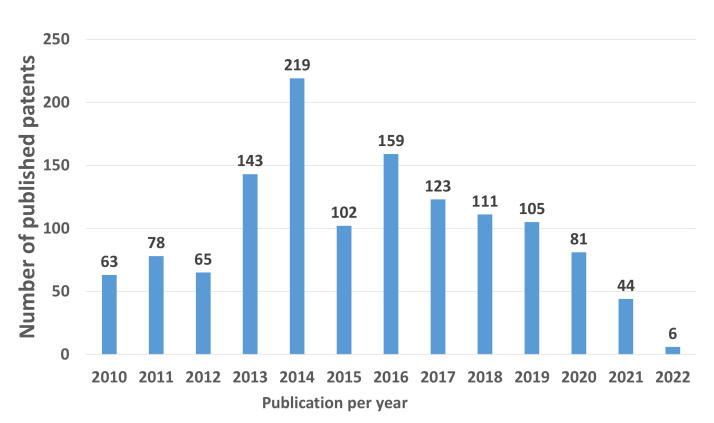
Annual number of published patents between 2010 and 2022.

**Figure 6 micromachines-13-00944-f006:**
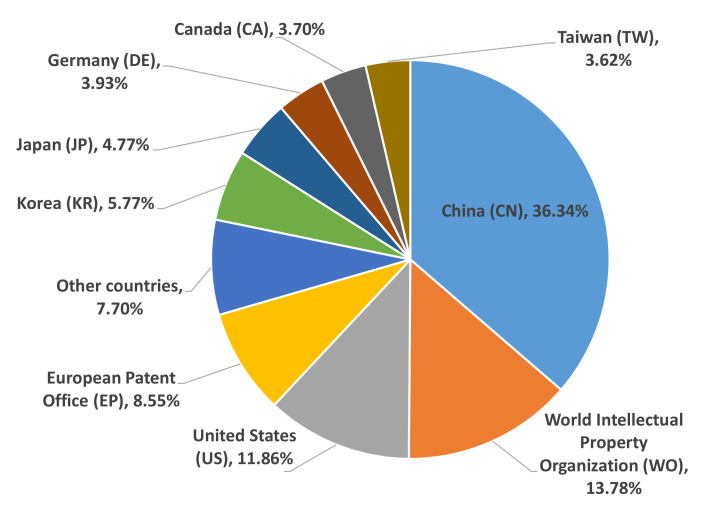
Distribution of origins of selected patents according to countries/assignees.

**Figure 7 micromachines-13-00944-f007:**
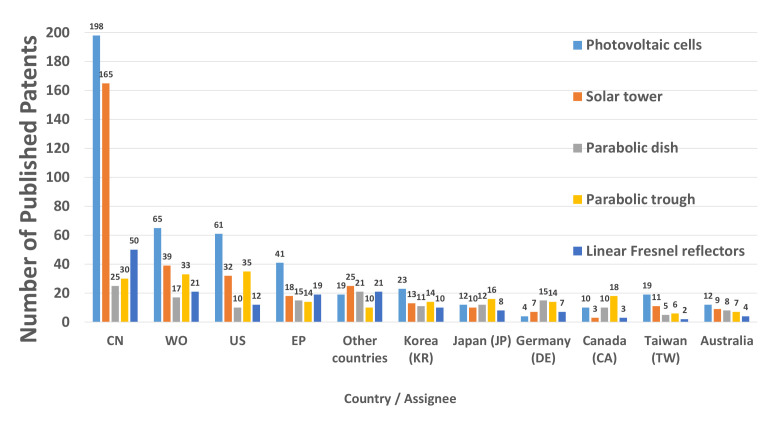
Number of patents and solar energy technologies in ORC application for different countries/organizations.

**Figure 8 micromachines-13-00944-f008:**
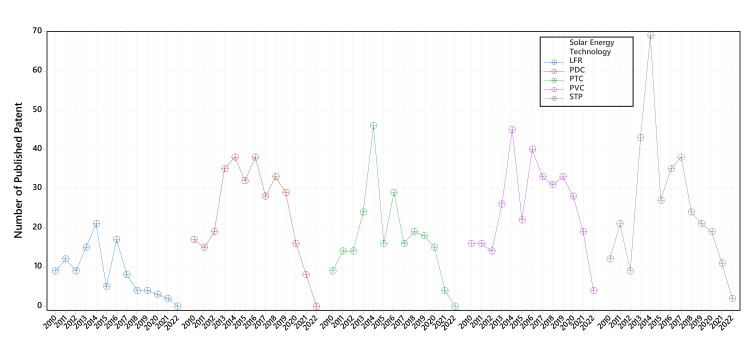
Patent publication pattern from January 2010 to May 2022.

**Figure 9 micromachines-13-00944-f009:**
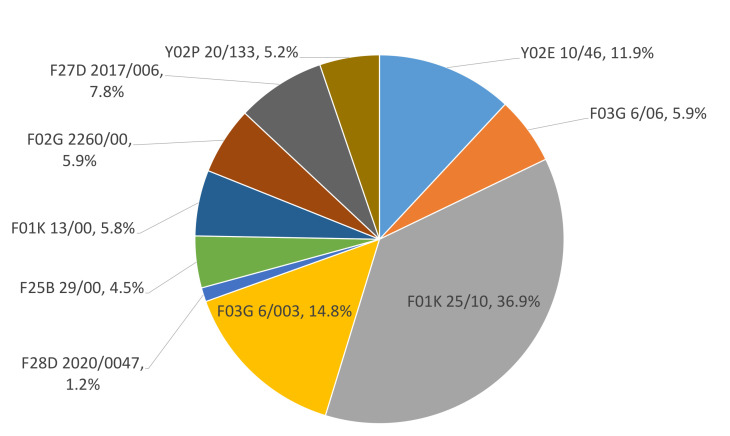
Patent distribution according to the IPC.

**Table 1 micromachines-13-00944-t001:** Strategy adopted to perform the search using google patents.

Technologies in ORC Application for Solar Energy	Keyword Terms
Parabolic trough	ORC and solar energy and parabolic trough (patent id, title, assignee, publication date)
Solar tower	ORC and solar energy and solar tower (patent id, title, assignee, publication date)
Linear fresnel collector	ORC and solar energy and linear fresnel collector (patent id, title, assignee, publication date)
Parabolic dish	ORC and solar energy and parabolic dish (patent id, title, assignee, publication date)
Photovoltaic cell	ORC and solar energy and photovoltaic cell (patent id, title, assignee, publication date)

**Table 2 micromachines-13-00944-t002:** IPC classification of selected patents.

Group	IPC	Subgroup
Steam engine plants, steam accumulators, engines using special working fluids (F01K) [42]	F01K25/10	Plants or engines characterized by use of special working fluids, not otherwise provided for; Plants operating in closed cycles and not otherwise provided for being cold
	F01K13/00	General layout or general methods of operation of complete plants
Spring, weight, inertia or like motors (F03G) [42]	F03G6/06	Devices for producing mechanical power from solar energy with solar energy concentrating means
	F03G6/003	Devices for producing mechanical power from solar energy having a Rankine cycle
Heat-exchange apparatus, not provided for in another subclass, in which the heat-exchange media do not come into direct contact (F28D) [42]	F28D2020/0047	Heat storage plants or apparatus in general, Regenerative heat-exchange apparatus not covered by groups using molten salts or liquid metals
Refrigeration machines, plants or systems, combined heating and refrigeration systems, heat pump systems (F25B) [42]	F25B29/00	Combined heating and refrigeration systems, e.g., operating alternately or simultaneously
Hot gas or combustion product positive displacement engine plants, use of waste-heat of combustion engines, not otherwise provided for (F02G) [42]	F02G2260/00	Recuperating heat from exhaust gases of combustion engine sand heat from cooling circuits
Details or accessories of furnaces, kilns, ovens, or retorts, in so far as they occur in more than one kind of furnace (F27D) [42]	F27D2017/006	Arrangements for using waste heat, Arrangements for using or disposing waste gases using a boiler
Reduction of greenhouse gas [ghg] emissions, related to energy generation, transmission or distribution (Y02E) [43]	Y02E10/46	Energy generation through renewable energy sources (Conversion of thermal power into mechanical power, e.g., Rankine, Stirling or solar thermal engines)
Climate change mitigation technologies in the production or processing of goods (Y02P) [43]	Y02P20/133	Technologies relating to chemical industry (Renewable energy sources, e.g., sunlight)

**Table 3 micromachines-13-00944-t003:** IPC codes not indexed in the initial search.

Group	IPC	Subgroup
Steam engine plants, steam accumulators, engine plants not otherwise provided for, engines using special working fluids or cycles [36]	F03G6/001	Devices for producing mechanical power from solar energy (having photovoltaic cells)
	F03G6/003	Devices for producing mechanical power from solar energy (having a Rankine cycle)
	F03G6/004	Devices for producing mechanical power from solar energy (of the Organic Rankine Cycle [ORC] type or the Kalina Cycle type)
	F03G6/02	Devices for producing mechanical power from solar energy (using a single state working fluid)
	F03G6/06	Devices for producing mechanical power from solar energy (with solar energy concentrating means)
	F03G6/061	Devices for producing mechanical power from solar energy (Parabolic trough)
	F03G6/062	Devices for producing mechanical power from solar energy (Parabolic dish)
	F03G6/063	Devices for producing mechanical power from solar energy (Tower concentrators)
	F03G6/065	Devices for producing mechanical power from solar energy (Parabolic dish)
	F03G6/066	Technologies relating to chemical industry (Renewable energy sources, e.g., sunlight)
Solar heat collectors, solar heat systems (for producing mechanical power from solar energy) [45]	F24S10/30	Solar heat collectors using working fluids (with means for exchanging heat between two or more working fluids)
Reduction of greenhouse gas [ghg] emissions, related to energy generation, transmission or distribution [46]	Y02E10/44	Energy generation through renewable energy sources (heat exchange systems)
	Y02E10/46	Energy generation through renewable energy sources (Conversion of termal power into mechanical power, e.g., Rankine, Stirling or solar termal engines)
	Y02E10/50	Energy generation through renewable energy sources (Photovoltaic energy)
	Y02E10/52	Energy generation through renewable energy sources (Photolvoltaic energy systems with concentrators)
Climate change mitigation technologies in the production or processing of goods [47]	Y02P20/133	Perfluorocarbons [PFC], Hydrofluorocarbons [HFC], Hydrochlorofluorocarbons [HCFC], Chlorofluorocarbons [CFC])
	Y02E20/155	Technologies relating to chemical industry (Renewable energy sources, e.g., sunlight)

## Data Availability

The data presented in this study are available on request from the corresponding author.

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
