# Peer review of "Solar Energy-Based Future Perspective for Organic Rankine Cycle Applications"

_micromachines, 2022, doi:10.3390/mi13060944_

Round 1

Reviewer 1 Report

Comments on micromachines-1750077

This study investigates the patents of solar energy technologies in the organic Rankine cycle (ORC) applications. The topic is interesting. However, the presentation of the manuscript is poor and needs to be improved. Some comments are given below.

(1)     Avoid the repeat information or sentence in the manuscript. For example, Figure 4 and Lines 176–185; Lines 276 and 278–279.

(2)     In Figure 1 and Lines 71–87, the scope of energy situation presented here is too wide. The authors should focus on the solar energy and ORC.

(3)     Equations 1 and 2 are not necessary to this study.

(4)     The same patent is generally filed in many offices or countries. However, in Line 171, the authors stated that “Cannot be registered in more than one country or assignee the name of a patent”, why?

(5)     Followed by the comment no. 4, the patent family is another important issue for the patent data analysis. However, no related information is given in the manuscript.

(6)     The postulation in Line 206–208 is questionable and not necessary.

(7)     In Lines 226–227, why the authors suddenly stated that “Collaboration projects between universities and the government will be necessary to promote use of ORC as a source of energy generation in a sustainable manner.”? This is incoherent with the content of the manuscript.

(8)     Because this study focuses on the patent data, section 4.3. Technology Updates were suggested to prepare from the filed patents, instead of the published papers.

(9)     Section 4.5. should be improved.

(10)  No related data and statement to Conclusion no. 5 were given in the manuscript.

(11)  Conclusion no. 6 is not the findings from the authors. It is well-known concept.

(12)  No related discussion to the Technology Life Cycle (Conclusion no. 7) was given in the manuscript.

Author Response

Thank you for your support, below are answers to the comments.

(1)     Avoid the repeat information or sentence in the manuscript. For example, Figure 4 and Lines 176–185; Lines 276 and 278–279.

Thanks for your observation. We have noted and made the changes shown below.

Figure 4 displays the patent selection process for this article using the PRISMA methodology [35]. The time-lapse analyzed is between January 2010 and May 2022.

Lines 176–185 were simplified for a better understanding of the idea. It was shortened to lines 174-178 and is shown below.

Of the initial sample of 3,510 patents, 743 were removed due to duplicate names being found. One thousand two hundred sixty-eight patents are eliminated because they are not directly related to an ORC. Additionally, 200 are terminated for filing active claims. Thus, the patents that meet the eligibility criteria and are part of the article are a total of 1,299.

Lines 276 and 278–279 were modified based on your insight and are shown as follows:

Figure 10 displays the IPC codes present in patents selected for this review. As can be seen, 77% of the patents are in group F. According to the EP; this group is called mechanical engineering, lighting, heating, weapons as well as blasting [43]. The remaining 23% is concentrated in group Y. As mentioned by EP [43], group Y is called general tagging of new technological developments; general tagging of cross-sectional technologies spanning over several sections of the IPC; technical subjects covered by former USPC cross-reference art collections [XRACs] and digests. Knowing the categories within the IPC makes it possible to find patents related to specific technologies quickly.

Table 2 presents a detailed description of the IPC codes that are part of the article. With the analysis of Table 2, it is easier to understand in which part of the IPC the patents are indexed.

Having IPC code at hand in many cases may not be feasible, and it is well-known that it is difficult to memorize the codes. In fact, with the discussion derived from Table 2, it is determined that there are patent codes on solar energy technologies in ORC applications that have not been described so far.

(2)     In Figure 1 and Lines 71–87, the scope of the energy situation presented here is too wide. The authors should focus on solar energy and ORC.

We appreciate your comment. The presentation of this section has been modified to define the context in which the article is developed.

Lines 71-87 were replaced by the following text:

An ORC is a thermodynamic system that allows production power or work and behaves in the same way as a cycle conventional Rankine to steam, the main difference is in the working fluid and the operating temperature of the cycle [9].

ORC system can generate power with temperatures up to 400 °C and capacities ranging from less than 1Kw to up to 10 MW. The implementation of solar energy in an ORC is a technology that can be applied in these plants due to the high compatibility between the operating temperatures of solar collection technologies and the operational temperature needs of the cycle [10].

The development and implementation of systems based on ORC with solar collection systems is a way to generate electricity in an environmentally friendly way. With this strategy, reducing greenhouse gases in the atmosphere is achieved, and sustainable technology to produce electricity is standardized [11].

Sun is the natural regulator of temperature in the terrestrial atmosphere. It can also radiate every day to our planet the energy consumed by society in one year. Thanks to the potential of this type of renewable energy and the technological development experienced in capturing solar radiation, the production of electricity from this type of energy has increased, as shown in Figure 1 [12].

(3)     Equations 1 and 2 are not necessary to this study.

Your comment has been analyzed and is considered correct. The equations have been removed.

(4)     The same patent is generally filed in many offices or countries. However, in Line 171, the authors stated that “Cannot be registered in more than one country or assignee the name of a patent”, why?

Thank you for your observation. As part of the eligibility process established in the Prisma methodology, it is established that there is no duplication in the name of the patents so that the total number of patents to be shown always refers to innovations that are different from each other.

(5)     Followed by comment no. 4, the patent family is another important issue for the patent data analysis. However, no related information is given in the manuscript.

Your comment is appreciated, for this reason, section 4.2 has been added to the article. The content of the section is modified, updating the content to present relevant patents published regarding solar technology for ORC applications.

(6)     The postulation in Line 206–208 is questionable and not necessary.

Your comment is correct. The sentence has been modified. 

Until May 1st, 2022, six patents had been published according to a search performed using Google Patents.

(7)     In Lines 226–227, why the authors suddenly stated that “Collaboration projects between universities and the government will be necessary to promote use of ORC as a source of energy generation in a sustainable manner.”? This is incoherent with the content of the manuscript.

Thank you for your comment. The sentence was removed.

(8)     Because this study focuses on the patent data, section 4.3. Technology Updates were suggested to prepare from the filed patents, instead of the published papers.

Your comment is precious to improve the quality of the article. Section 4.3 has been modified based on his proposal. The focus of the section was changed so that relevant patents are presented instead of articles.

(9)     Section 4.5. should be improved. 

Thank you for your comment. Section 4.5 has been improved based on your advice. In addition, articles with different views on patent citations have been added, and finally, we have offered our perspective.

(10)  No related data and statement to Conclusion no. 5 were given in the manuscript.

Your comment is completely correct. Conclusion no. 5  was removed.

(11)  Conclusion no. 6 is not the findings of the authors. It is a well-known concept.

Your comment is completely correct. Conclusion no. 6  was removed.

(12)  No related discussion of the Technology Life Cycle (Conclusion no. 7) was given in the manuscript.

Your comment is completely correct. The no. 7 conclusion was removed.

Reviewer 2 Report

The paper analyzes in detail the development of organic Rankine cycle power generation technology using solar energy technology, especially the related patents in recent years. This is an important and interesting topic. The paper makes a reasonable classification and compares the technological development of different countries and regions. The writing of the paper is smooth and the logic is clear. It can be seen that the authors are also very attentive in the production of the Figures, and they are clear and easy to read. So my opinion is that this is a good review paper and should be accepted for publication.

Author Response

We appreciate your comment, we hope the manuscript will be published soon.

Reviewer 3 Report

1-Please update the reference. According to Google Scholar, to date, there are 2150 published articles. Yet, out of 85 reference, only less than 10 have been cited.

2-Introduction part must included more 2022 and 2021 articles.

3-More updated data must be included such as Figure 1 and Figure 2. Why only 2019?

Author Response

1-Please, update the reference. According to Google Scholar, to date, there are 2150 published articles. Yet, out of 85 references, only less than 10 have been cited.

Thank you for your comment. The references were updated. Due to the focus of the article, many patents are referenced, although the bibliography cited from the articles is up to date. 

2-Introduction part must include more 2022 and 2021 articles.

Thank you for your comment. Some references were modified, and information from articles with greater validity has been cited. Based on your recommendation, a total of 4 references have been added.

3-More updated data must be included such as in Figure 1 and Figure 2. Why only 2019?

Thank you for your comment. The theme of the figures and the relevance and timeliness of the information shown have been reviewed. As a result, table 2 was removed, and Table 1 has been updated.

Round 2

Reviewer 1 Report

Most comments given previously were replied in the revision. No more comments now.

Reviewer 3 Report

N/A